# The Impact of Lower-Strength Alcohol Products on Alcohol Purchases by Spanish Households

**DOI:** 10.3390/nu14163412

**Published:** 2022-08-19

**Authors:** Peter Anderson, Daša Kokole

**Affiliations:** 1Care and Public Health Research Institute, Maastricht University, 6200 MD Maastricht, The Netherlands; 2Population Health Sciences Institute, Newcastle University, Baddiley-Clark Building, Newcastle upon Tyne NE2 4AX, UK

**Keywords:** no-alcohol products, low-alcohol products, Spain, household purchase data, Kantar WorldPanel

## Abstract

In its action plan (2022–2030) to reduce the harmful use of alcohol, the WHO calls on economic operators to “substitute, whenever possible, higher-alcohol products with no-alcohol and lower-alcohol products in their overall product portfolios, with the goal of decreasing the overall levels of alcohol consumption in populations and consumer groups”. In this paper, we investigate substitution at the level of the consumer based on Spanish household purchase data. ARIMA modelling of market research data of 1.29 million alcohol purchases from 18,954 Spanish households is used to study the potential impact of lower-strength alcohol products on reducing household purchases of grams of alcohol between the second quarter of 2017 and the first quarter of 2022. For households that recently bought either no-alcohol beer or wine (ABV ≤ 0.5%), the subsequent associated purchases of higher-strength beers and wines, respectively, and total grams of alcohol were reduced, the more so the higher the volume of initial purchases of beers and wines. The introduction of 20% ABV variants of same-branded 40% ABV whisky and gin during early 2021 was associated with reduced purchases of grams of alcohol within all spirits and of total grams of alcohol as a result of switching from other spirits products to the 20% variants, although not associated with reduced purchases of grams of alcohol within all variants of the studied same-branded whisky and gin; instead, an increase was observed in this category. With respect to Spanish household purchase data, the evidence behind the WHO’s call for substitution is substantiated. Further research across different jurisdictions is needed to provide ongoing monitoring of the impact of potential substitution on consumer behavior and public health, including unintended consequences, with findings from research informing future alcohol policies at all levels.

## 1. Introduction

Alcohol is toxic to health [1], is genotoxic, and is a carcinogen [2,3,4,5]. Forty-year-olds who drink more than 350 g of alcohol a week (about five drinks a day) lose four to five years of life compared to those who drink 100 g of alcohol or fewer a week (about one-and-a-half drinks a day) [6]. The key to reducing the harm done by alcohol is to drink less alcohol [7]. The WHO SAFER [8] initiative calls on governments to encourage people to drink less alcohol by making alcohol more expensive; decreasing its availability; banning or strictly regulating its advertising; putting in place strict drink-driving laws; and providing advice, support, and treatment to heavy drinkers, all of which are effective in reducing alcohol consumption [9].

Furthermore, in its action plan (2022–2030) to “effectively implement the global strategy to reduce the harmful use of alcohol as a public health priority”, the WHO calls on economic operators to “substitute, whenever possible, higher-alcohol products with no-alcohol and lower-alcohol products in their overall product portfolios, with the goal of decreasing the overall levels of alcohol consumption in populations and consumer groups, while avoiding the circumvention of existing regulations for alcoholic beverages and the targeting of new consumer groups with alcohol marketing, advertising and promotional activities” [10]. The theory of change behind this call is that at the level of the consumer, the substitution of higher-alcohol products with no-alcohol and lower-alcohol products leads to less alcohol consumed and, thus, to less harm done by alcohol [11].

Based on British household purchase data, we previously demonstrated that at the level of the consumer, substitution of higher-strength beers with lower-strength beers occurs, leading to an overall decrease in purchases of grams of alcohol [12,13,14]. Households that had previously bought same-branded regular-strength beers and that went on to buy newly-introduced same-branded no- and low-alcohol beers subsequently reduced purchases of regular-strength beers by 48.5 mL per adult per household per day for days in which a purchase was made, representing a 22.5% reduction and matched by new purchases of 34.6 mL of the new no- and low- alcohol beers, with such changes remaining stable over at least two years of follow-up (the length of time available for analyses) [12]. The introduction of 46 new no- and low- (ABV ≤ 3.5%) alcohol beers and the reformulation of 33 existing beers to contain less alcohol (of 1903 available beer brands) were associated with reductions in purchases of total grams of alcohol across all households, with a larger reduction associated with reformulation (3.9%) than with the introduction of new no- and low-alcohol beer (2.6%) [13]. For every 10 mL increase in purchases of no-alcohol beer per adult per household per day, purchases of total grams of alcohol contained within beer dropped by 1.1% [14]. For every drop in the absolute value of the ABV of beer of 0.1% over time (from a baseline of 4.34), purchases of total grams of alcohol contained within beer dropped by 6.9% [14], and for every 5 mL increase in purchases of no-alcohol wine products per adult per household per day, purchases of total grams of alcohol contained within wine dropped by 1.2% [14]. However, the drawback of the cited analyses is that they are based on just one jurisdiction, Great Britain. In this paper, we examine the theory of change based on analyses of Spanish household purchase data for the period from the second quarter of 2017 to the end of the first quarter of 2022.

Spain is one of the European countries with the highest production levels of no-alcohol beer (alcohol by volume, ABV ≤ 0.5%, as defined by European Commission) [15]; we therefore consider whether, at the level of the consumer, there is substitution within beers from higher-strength beers (ABV > 0.5%) to no-alcohol beers and whether this results in fewer grams of alcohol purchased. Because in Spain, no-alcohol beer has a different definition (ABV ≤ 1.0%) than in other jurisdictions [16], we also assess substitution at this level. Spain is a traditional wine-consuming country [17]; therefore, we consider whether, at the level of the consumer, there is substitution within wines from higher-strength wines (ABV > 0.5%) to no-alcohol wines (ABV ≤ 0.5%) and whether this results in fewer grams of alcohol purchased, noting that due to global heating [18,19], the ABV of wines is, in general, increasing [20,21,22]. At the beginning of 2021, two lower-alcohol-strength spirit-based products (one variant of whisky and one variant of gin) were launched in Spain: 20% ABV variants of same-branded regular-strength whisky and gin (40% ABV) [23]. To the best of our knowledge, these are the only two examples of same-branded lower-strength spirit products introduced during the time period of the present study; therefore, we consider whether, at the level of the consumer, there is substitution from the higher- to the lower-strength variants and whether this results in fewer grams of alcohol purchased.

The specific hypotheses to be tested are:The purchase of no-alcohol beers is substituted for the purchase of all other higher-strength beers, leading to fewer grams of alcohol purchased;The purchase of no-alcohol wines is substituted for the purchase of all other higher-strength wines, leading to fewer grams of alcohol purchased; andThe purchase of 20% ABV variants of same-branded whisky and gin is substituted for the purchase of the regular-strength variants of the whisky and gin, leading to fewer grams of alcohol purchased.

## 2. Methods

### 2.1. Study Design

We use interrupted time-series analyses to investigate the potential impact of consumers purchasing no-alcohol or lower-alcohol products as a substitute for higher-strength products, leading to reduced purchases of grams of alcohol, the primary outcome.

### 2.2. Data Source

Our data source was Kantar Worldpanel’s (KWP) household shopping panel. KWP comprises approximately 12,000 Spanish households at any one time, recruited via stratified sampling, with targets set for province, household size, and age of the main shopper; the panel is representative of households in Spain as a whole. Households provide demographic information when joining the panel, followed by annual updates and quality checks. Using barcode scanners, households record all alcohol purchases brought into the home from all store types, including Internet shopping.

We analyzed raw KWP data on take-home purchases of alcohol products in Spain for the time period from the second quarter of 2017 to the end of the first quarter of 2022. For each individual purchase, the provided data included the type and volume of the purchase, the brand, and the alcohol content by volume (ABV). For one-third of purchases, a band of ABV was provided rather than the specific ABV; for these products, the mid-value of the ABV band was taken. The provided data categorized each purchase as either a beer, a wine, a sparkling wine, or a spirit-based product. The volume purchased was combined with ABV to calculate grams of alcohol purchased. Households were grouped by age of the main shopper, social grade, and autonomous community (Appendix A).

We categorized data, first, for any day that a household bought alcohol, summing the amount of alcohol purchased in both volume and grams, divided by the number of adults in the household. Then, for each day of the study period, we calculated the number of alcohol purchases, the mean volume and grams of purchases, and the mean alcohol by volume of purchases across all households for all products and for each of three categories of products: beers, wines, and spirits.

For beer and wine, we individually tagged those households that had purchased beer/wine with an ABV > 0.5% and that had purchased beer/wine with an ABV ≤ 0.5%, with the first purchase of the beer/wine with an ABV ≤ 0.5% occurring at least three months (and, for sensitivity analysis, at least six months) since the date of the first purchase of beer/wine with an ABV > 0.5% to identify households newly buying no-alcohol beer/wine. For beer, we also individually tagged those households that had purchased beer with an ABV > 1.0% and that had purchased beer with an ABV ≤ 1.0%, with the first purchase of beer with an ABV ≤ 1.0% occurring at least three months since the date of the first purchase of beer with an ABV > 1.0%. We also individually tagged those households that had purchased any variant of the whisky or gin, dividing them into those that had or had not made at least one purchase of a 20% variant of the whisky or gin.

## 3. Statistical Analyses

For all households, we plotted the number of purchases over time separately for beers, wines, and the studied same-branded whisky and gin. Because the calculations of volumes and grams purchased are based on days when a household made an alcohol purchase, as in our previous studies on British household purchase data [17,18,19,20], we calculated and described frequencies of purchases at the household level as the number of days between any one purchase and the next purchase.

For hypotheses 1 and 2, (the purchases of no-alcohol beers/wines substituted for the purchase of higher-strength beers/wines, leading to fewer grams of alcohol purchased), we select two data files, one for beer and one for wine, including, households that had made at least one purchase of no-alcohol beer/wine and at least one purchase of any other beer/wine, with the first purchase of the no-alcohol beer/wine occurring more than 90 days after the first purchase of any other beer/wine, categorizing such households new purchasers of no-alcohol beer/wine.

For each household, we set the day of the first purchase of the no-alcohol beer/wine as day 1, i.e., the event, numbering all other days as before (minus days) or after (plus days). We undertook interrupted time-series analyses to investigate the associated impact of the event (the first day of purchase of the no-alcohol variant) on changes in the total purchased grams of alcohol (the primary outcome) and on changes in the total volume of other beer/wine purchased (the secondary outcome). Thus, the dependent variables were the number of grams of alcohol purchased and the volume of purchases of all beer/wine with an ABV >0.5%. The independent variables were the event (the first day of purchase of no-alcohol beer/wine) and alcohol by volume of all beer/wine with an ABV >0.5% to control for any changes in ABV as a measure of potential reformulation of higher-strength products over time. (For details, see Appendix A).

We undertook two sensitivity analyses: first, we repeated the analyses, having set the first purchase of the no-alcohol beer/wine occurring more than 180 days after the first purchase of any other beer/wine; then, for beer, we repeated the analysis using 1.0% (as opposed to 0.5%) ABV as the division between no-alcohol beer and all other beer.

We repeated the models to investigate whether sociodemographic variables impacted the associated change according to each sociodemographic variable (age of main shopper, social grade of household, and autonomous community in which the household is located), including as additional independent variables the sociodemographic group, (Appendix A) and the interaction term sociodemographic group*event.

For all analyses, we present the unstandardized coefficients and 95% confidence intervals (CIs) arising from the interrupted time-series regression models, and we plotted the predicted values of the dependent variables from the interrupted time-series regression models.

For Hypothesis 3, (the purchases of 20% ABV variants of the same-branded whisky and gin as a substitute for the purchase of regular-strength variants of the whisky and gin (ABV = 40%), leading to fewer grams of alcohol purchased), we restrict the analyses to households that had made any purchases of any variant of the whisky or gin, dividing the households into those that had or had not made at least one purchase of a 20% ABV variant.

We set the event as the introduction of 20% ABV variants, measured as the first recorded day of any purchase of whisky or gin with an ABV of 20% by any household. We examined changes for those households that had made any purchase of 20% variants, using changes amongst households that had made at least one purchase of the regular-strength but not the ABV 20% variants as controls. We used interrupted time-series analyses to examine associated level changes for the following dependent variables: the total grams of alcohol purchased within all spirits as the primary outcome; and, as secondary outcomes: the volume of purchases of all regular-strength variants, the total grams of alcohol purchased within any variant of whisky and gin, the total grams of alcohol purchased within any spirits other than the branded whisky or gin (to examine potential shifts from other spirit products to the 20% variants), the total grams of all alcohol purchased, and the ABV of all spirits purchased. The independent variable was the event (the date of first purchase by any household of an ABV 20% variant). To control for the changes amongst households that had made at least one purchase of the regular-strength variants but not ABV 20% variants, we added to the model ‘whether or not’ the household had bought a 20% variant and the interaction term ‘whether or not’*event (see Appendix A for details).

We present the unstandardized coefficients and 95% confidence intervals (CIs) arising from the interrupted time-series regression models and plotted the predicted values of the dependent variables arising from the interrupted time-series regression models.

Given the small number of purchases of the 20% variants, we did not repeat the models to investigate whether sociodemographic variables impacted the associated level changes.

All analyses were performed with SPSSv27 [24].

## 4. Results

### Households and Purchases

We analyzed data from 18,954 Spanish households with 1.29 million separate alcohol purchases between the beginning of the second quarter of 2017 to the end of the first quarter of 2022. The mean frequency of purchases of any alcohol product (calculated as the number of days between one purchase and the next) was 21.2 days (95% CI = 21.1 to 21.3), with this value remaining stable throughout the follow-up period (regression coefficient, frequency with days of the time period = −1.3^−4^; 95% CI = −3.0^−4^ to 0.4^−4^).

The number of purchases of beers, wines, and branded whisky/gin are plotted in Appendix A. Overall, 12.4% of all beer purchases and 3.9% of all wine purchases had an ABV ≤0.5%, with the trends for no-alcohol beer remaining stable over time and the trends for no-alcohol wine decreasing slightly over time (see Appendix A). For beer, 20.5% of all purchases had an ABV ≤ 1.0%. Since the time of introduction of the 20% whisky/gin variants, out of 648 purchases of all variants, 49 (7.6%) were purchases of the 20% variants.

**Hypothesis** **1** **(H1)**.*purchases of no-alcohol beers substituted for the purchase of higher-strength beers, leading to fewer grams of alcohol purchased*.

For households newly purchasing no-alcohol beer (*n* = 4115), accounting for 21.7% of all households, the event, i.e., the start of new purchases of no-alcohol beer, was associated with decreased purchases of all other beer and a drop in purchases of total grams of all alcohol of 5.3 g (95% CI = 5.0 to 5.7) (per adult per household per day of purchase; Table 1), which remained stable over time (Figure 1, top graph). The first sensitivity analysis, with the first purchase of the no-alcohol beer occurring more than 180 days after the first purchase of any other beer (*n* = 3227 households, 17.0% of all households) provided similar results (Table 1). The second sensitivity analysis, with the definition of no-alcohol beer as an ABV ≤1.0%, also provided similar results, although in this case, due to the higher cut-off of ABV, the associated reduction in overall levels of alcohol purchased was lower, at 2.4 g (95% CI = 1.2 to 2.8).

The associated reduction in purchased grams of alcohol was greater amongst younger than older households but showed no clear pattern according to social grade (Table 2). With respect to the variable of autonomous community, associated reductions in purchased grams of alcohol were greater when higher volumes of beer were purchased prior to the event (Table 2). The standardized regression coefficient of the associated reduction with the initial volume of beer purchased prior to the event was −0.735 (95% CI = −0.960 to −0.313).

**Hypothesis** **2** **(H2)**.*purchases of no-alcohol wines substituted for the purchase of higher-strength wines, leading to fewer grams of alcohol purchased*.

For households newly purchasing no-alcohol wine (*n* = 1271), the event, i.e., the start of new purchases of no-alcohol wine, was associated with decreased purchases of all other wine and a drop in purchases of grams of all alcohol of 8.2 g (95% CI = 7.8 to 8.6) (per adult per household per day of purchase, Table 1), with this value remaining stable over time (Figure 1, bottom graph). Sensitivity analysis, with the first purchase of the no-alcohol wine occurring more than 180 days after the first purchase of any other wine (n households = 977) provided similar results (Table 1). The associated reduction in purchased grams of alcohol was greater amongst older than younger households and amongst those with lower rather than higher social grades (Table 2). With respect to the variable of autonomous community, associated reductions in purchased grams of alcohol were greater when higher volumes of wine were purchased prior to the event (Table 2). The standardized regression coefficient of the associated reduction with the initial volume of wine purchased prior to the event was −0.618, (95% CI = −1.432 to −0.253).

**Hypothesis** **3** **(H3)**.*purchases of lower-strength spirits substituted for the purchase of higher-strength spirits, leading to fewer grams of alcohol purchased*.

For households that had made at least one purchase of the 20% ABV variants of whisky or gin, the event, i.e., the introduction of the 20% ABV variants, as evidenced by the date of the first household purchase, controlling for any changes in purchases of the regular-strength variants by households that did not purchase the 20% ABV variants, was associated with a reduction of 26.7 g of alcohol in all spirit products (95% CI = 23.6 to 29.8) per adult per household per day of purchase, with the mean % ABV of purchased spirits decreasing by 6.1 (95% CI = 5.8 to 6.4) (Table 3; Figure 2, top graph). This reduction seemed to occur in the context of switching purchases from other spirit products to the 20% variants. The event was associated with purchases of the 20% variants of 0.96 bottles per adult per household per day of purchase (95% CI = 0.95 to 0.97), there was no drop but a small increase in purchases of the regular-strength variants of, on average, 0.07 bottles per adult per household per day of purchase (95% CI = 0.05 to 0.10) (Figure 2, middle graph), whereas this was associated with a decrease in purchases of grams of alcohol in all products of 8.7 g per adult per household per day of purchase (95% CI = 1.7 to 15.8) (Figure 2, middle graph), and there was an associated increase in purchases of 22.2 g of alcohol contained within all variants of the same-branded whisky or gin (95% CI = 21.5 to 22.8) (Figure 2, bottom graph) but an associated decrease in purchases of all other spirits of 41.9 g per adult per household per day of purchase (95% CI = 40.9 to 43.0).

## 5. Discussion

Based on analyses of KWP Spanish household purchase data between 2017 (from the second quarter) to 2022 (end of the first quarter), our three hypotheses are confirmed. For households that newly bought either no-alcohol beer with an ABV ≤ 0.5% (Hypothesis 1) or no-alcohol wine with an ABV ≤ 0.5% (Hypothesis 2), the subsequent associated purchases of higher-strength beers and wines, respectively, and of total grams of alcohol were reduced. At the household level, we set 90 days as the time between the first purchase of a higher-strength product and the first purchase of a no-alcohol product to define a new no-alcohol purchase; the sensitivity analyses with the time set at 180 days provided similar results. We used the European Commission definition of a no-alcohol beer (ABV ≤ 0.5%); sensitivity analyses using the Spanish definition of 1.0% ABV resulted in similar findings, although, due to the higher-ABV definition of no-alcohol beer, smaller reductions were observed for the associated decrease in purchased grams of all alcohol. When examining the associated changes according to the autonomous communities in which the households were located, the higher the volume of initial purchases of beers and wines, the greater the associated reductions in total grams of alcohol purchased.

For spirits and with respect to Hypothesis 3 (purchases of 20% ABV variants of same-branded whisky and gin substituted for the purchase of regular-strength variants, leading to fewer grams of alcohol purchased), we took a different approach. We examined changes resulting from the unique introduction of 20% ABV variants of existing regular-strength variants. Given the small number of purchases, we used purchases of grams of alcohol within all spirits as the main outcome and did not examine changes according to sociodemographic household groups. We did not observe substitution within the specific brands but found reductions in associated purchases of grams of all spirits with switching from other higher-strength spirits products to the 20% variants of whisky and gin, confirming Hypothesis 3.

Overall, the percentages of the studied no-alcohol products, particularly beer, as a proportion of regular-product purchases were relatively high, indicating an openness of consumers to buying no-alcohol products in Spain, although the trend was stable for the past few years, as opposed to increasing. Despite this, substitution occurs in only a relatively low proportion of households (one-fifth for beer and even less for wine); thus, although individuals may experience improved health due to reduced alcohol consumption, the impact at the population level will be limited.

A strength of the present study is that we included a large number of alcohol purchases from a large number of households, with a large number of data points before and after the examined events and scanned barcode data providing objective data. Examining household purchases in Spain, with a drinking culture that differs considerably from that of Great Britain, we found evidence for substitution and reduced purchases of grams of alcohol—not only with respect to the impact of no-alcohol beers and wines but also related to the impact of reduced-alcoholic-strength variants of spirit products. With respect to beers and wines, the associated reductions occurred across all age and social-grade groups, and when examined according to autonomous community, these reductions were higher with higher initial volume of purchases.

A main limitation of the present study is that we were only able to assess changes in household alcohol purchases from shops and supermarkets and excluded alcohol consumed in cafés, bars, and restaurants. Furthermore, we only examined purchases and not actual levels of alcohol consumption for the time periods studied. Adults in a household may not be responsible for an equal share of the alcohol purchased, and not all adults in a household are necessarily drinkers. The data are also subject to limitations, with heavy drinkers tending to be under-represented in household panel data [25] and with alcohol purchases tending to be under-reported in these datasets [26,27].

## 6. Conclusions

From the perspective of household purchases, the theory of change, i.e., that substitution of higher-alcohol products for no-alcohol and lower-alcohol products leads to less alcohol consumed, behind WHO’s calls to economic operators to “substitute, whenever possible, higher-alcohol products with no-alcohol and lower-alcohol products in their overall product portfolios, with the goal of decreasing the overall levels of alcohol consumption in populations and consumer groups” is substantiated by the Spanish dataset, as previously confirmed by the British dataset [12,13,14]. In a previous study, we showed that pricing policy, in particular minimum unit price, can be set to favor substitution [28]. There is a need for expanded research across different jurisdictions to monitor the impact of potential substitution on consumer behavior and public health, including unintended consequences, with findings from research informing future alcohol policies at all levels. Substitution is an addition to, and not a replacement of, the elements of the WHO’s SAFER initiative that need to be implemented to reduce the harmful use of alcohol [13], ensuing that economic operators in alcohol production and trade, as called for by the WHO, “abstain from interfering with alcohol policy development and refrain from activities that might prevent, delay or stop the development, enactment, implementation and enforcement of high-impact strategies and interventions to reduce the harmful use of alcohol” [15].

## Figures and Tables

**Figure 1 nutrients-14-03412-f001:**
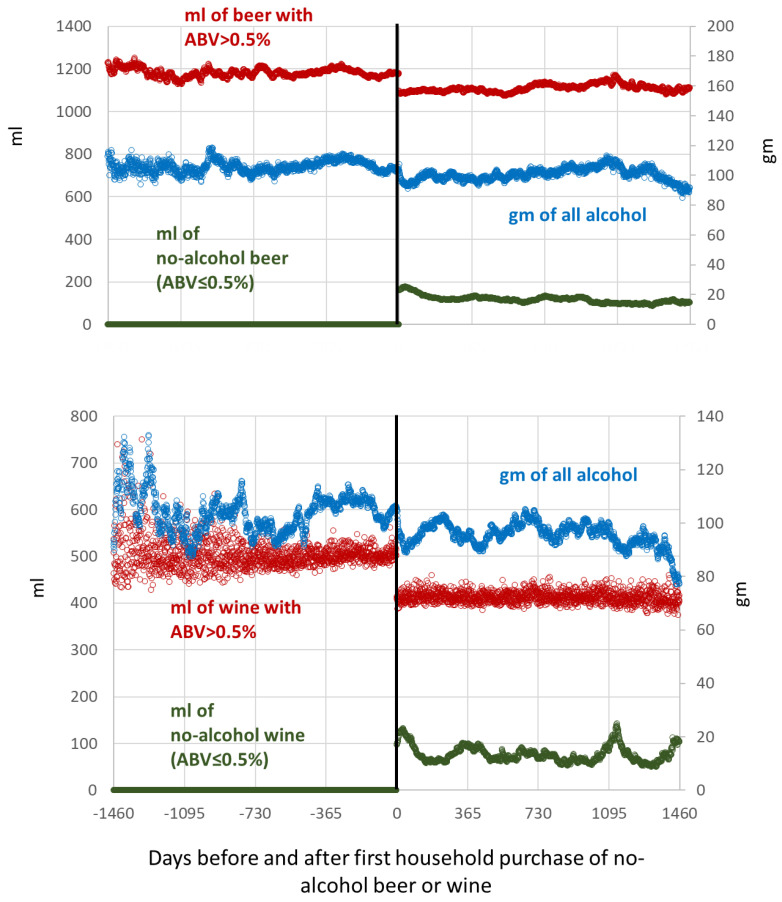
Households that had bought at least one no-alcohol beer and at least one other-strength beer (**Top graph**), with the first purchase of no-alcohol beer >3 months after the first purchase of other-strength beer (new purchasers of no-alcohol beer) (*n* = 4115 households). Households that had bought at least one no-alcohol wine and at least one other-strength wine (**bottom graph**), with the first purchase of no-alcohol wine >3 months after the first purchase of other-strength wine (new purchasers of no-alcohol wine) (*n* = 1271 households). **Left axis**: mL purchased per adult per household per day of purchase per study day; **right axis**: grams of all alcohol purchased per adult per household per day of purchase per study day; **vertical line**: first household purchase of no-alcohol beer or wine; **data points**: daily, with exponential smoothing.

**Figure 2 nutrients-14-03412-f002:**
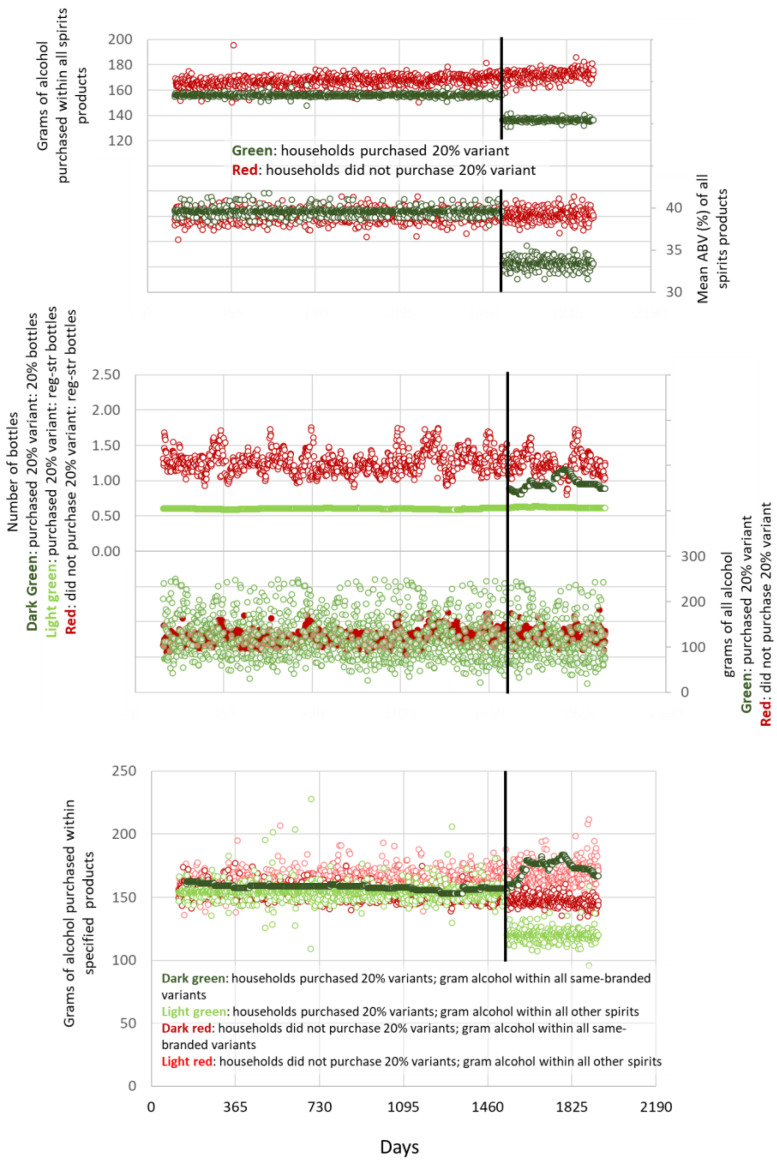
Households that had bought at least one variant of the same-branded whisky or gin, divided into those that had bought the newly introduced 20% variant (green) and those that had not (red). **Top graph:** total grams of alcohol purchased per day of purchase per study day within any spirit product and mean ABV of any purchased spirit product. **Middle graph**: number of bottles of whisky and gin purchased (adjusted to 700 mL and smoothed per week) and total grams of alcohol purchased per day of purchase per study day. **Bottom graph**: total grams of alcohol purchased within any variant of whisky and gin and within any other spirit product per day of purchase per study day. **Vertical line**: introduction of 20% variant, as identified by first household purchase. **Data points**: daily, with exponential smoothing.

**Table 1 nutrients-14-03412-t001:** Unstandardized coefficients (95% CI) from interrupted time-series regression models for beer and wine. For beer and wine, first purchase >180 days: sensitivity analysis 1. For beer, no-alcohol beer, defined as ≤1.0%: sensitivity analysis 2.

		Intercept		Level Change Due to Event		ABV All Other Beer/Wine	
		First Purchase > 90 Days	First Purchase > 180 Days	First Purchase > 90 Days	First Purchase > 180 Days	First Purchase > 90 Days	First Purchase > 180 Days
Beer: no-alcohol beer with ABV ≤0.5%	No-alcohol beer (mL)	0	0	116.5 (115.6 to 117.4)	113.7 (112.9 to 114.5)	-	-
	All other beer (mL)	1181.0 (1180.0 to 1182.0)	1181.9 (1180.8 to 1183.0)	−72.7 (−74.1 to −71.4)	−71.9 (−73.4 to −70.3)	-	-
	Alcohol in all products (grams)	105.0 (102.4 to 107.6)	108.2 (105.7 to 110.7)	−5.3 (−5.7 to −5.0)	−3.9 (−4.2 to −3.6)	0.14 (−0.40 to 0.67)	−0.43 (−0.94 to 0.08)
Beer: no-alcohol beer with ABV ≤1.0%	No-alcohol beer (mL)	0	-	150.0 (149.4 to 150.6)	-	-	-
	All other beer (mL)	1204.3 (1203.2 to 1205.5)	-	−114.6 (−116.2 to −113.0)	-	-	-
	Alcohol in all products (grams)	118.9 (114.6 to 123.3)	-	−2.4 (−2.8 to −2.1)	-		
Wine	No-alcohol wine (mL)	0	0	77.2 (76.2 to 78.2)	79.8 (79.6 to 80.0)	-	-
	All other wine (mL)	505.0 (503.5 to 506.4)	508.1 (508.0 to 508.2)	−92.1 (−94.1 to −90.1)	−96.6 (−96.4 to −96.8)	-	-
	Alcohol in all products (grams)	104.3 (101.1 to 107.5)	103.2 (100.8 to 105.6)	−8.2 (−8.6 to −7.8)	−8.2 (−8.6 to −7.9)	−0.06 (−0.33 to 0.21)	0.12 (−0.09 to 0.32)

**Table 2 nutrients-14-03412-t002:** Unstandardized coefficients (95% CI) from interrupted time-series regression models according to sociodemographic variables for beer and wine. The model includes, as independent variables, the event, the sociodemographic category, and the interaction term event*sociodemographic category. Thus, the coefficients for the level changes represent the differences in changes between the stated category and the reference category.

	Beer			Wine	
Age	Intercept	Level Change		Intercept	Level Change
18–34	113.6 (112.9 to 114.4)	−28.53 (−29.37 to −27.69)		118.9 (117.3 to 120.5)	−7.15 (−8.55 to −5.75)
35–49	113.3 (113.0 to 113.5)	−10.63 (−11.42 to −9.83)		111.7 (111.5 to 111.9)	−5.13 (−6.36 to −3.90)
50–64	92.6 (92.3 to 92.9)	2.58 (1.79 to 3.38)		86.6 (86.2 to 87.0)	12.83 (11.59 to 14.06)
65+ (reference category)	114.8 (114.6 to 115.0)	0.00 (. to .)		119.0 (118.7 to 119.4)	0.00 (. to .)
**Social Grade**					
Medium–high and high	104.5 (104.2 to 104.9)	2.37 (1.60 to 3.15)		87.3 (86.5 to 88.1)	31.34 (30.10 to 32.58)
Medium–medium	96.2 (96.0 to 96.4)	−3.62 (−4.40 to −2.85)		106.9 (106.0 to 107.8)	−0.34 (−1.55 to 0.87)
Medium–low	105.7 (105.3 to 106.2)	7.45 (6.68 to 8.23)		110.4 (110.0 to 110.7)	5.62 (4.41 to 6.83)
Low (reference category)	113.6 (113.1 to 114.1)	0.00 (. to .)		107.6 (107.2 to 107.9)	0.00 (. to .)
**Autonomous Community**					
Castilla–La Mancha ^a^	134.1 (133.9 to 134.4)	−46.0 (−47.0 to −45.1)	Galicia ^a^	146.4 (143.7 to 149.1)	−92.5 (−94.7 to −90.4)
Balearic Islands	119.9 (119.2 to 120.5)	−27.4 (−28.4 to −26.5)	Catalonia	143.4 (142.9 to 143.9)	−71.9 (−73.8 to −69.9)
Catalonia	117.4 (116.9 to 117.9)	−31.7 (−32.6 to −30.8)	Castilla–La Mancha	138.4 (137.9 to 138.9)	−37.5 (−39.5 to −35.5)
Aragon	111.8 (111.8 to 111.8)	−29.7 (−30.6 to −28.8)	Navarre	134.7 (134.7 to 134.7)	−19.4 (−23.0 to −15.8)
Valencian Community	108.4 (108.2 to 108.6)	−25.1 (−26.0 to −24.2)	Castile and León	111.4 (111.1 to 111.6)	−11.4 (−13.4 to −9.4)
Castile and León	106.5 (106.5 to 106.5)	−14.4 (−15.3 to −13.5)	Community of Madrid	102.2 (101.9 to 102.5)	−36.7 (−38.6 to −34.7)
Extremadura	106.3 (105.8 to 106.8)	−13.6 (−14.5 to −12.7)	Andalusia	100.4 (100.3 to 100.6)	−28.3 (−30.2 to −26.4)
Andalusia	105.2 (104.9 to 105.5)	−14.7 (−15.6 to −13.8)	La Rioja	98.4 (98.4 to 98.4)	−51.3 (−56.3 to −46.3)
Canary Islands	101.9 (101.5 to 102.3)	−37.8 (−38.7 to −36.9)	Basque Country	94.4 (91.1 to 97.7)	57.5 (55.0 to 60.0)
Basque Country	98.4 (98.1 to 98.6)	−14.3 (−15.2 to −13.4)	Murcia	94.2 (93.4 to 95.0)	−18.9 (−21.2 to −16.7)
Community of Madrid	96.0 (95.7 to 96.3)	−20.7 (−21.6 to −19.8)	Aragon	89.4 (89.4 to 89.4)	−19.4 (−21.9 to −16.9)
Galicia	95.0 (94.7 to 95.4)	−25.3 (−26.2 to −24.4)	Balearic Islands	86.1 (83.8 to 88.3)	−19.4 (−21.7 to −17.2)
Murcia	91.8 (90.9 to 92.7)	−13.6 (−14.5 to −12.7)	Extremadura	82.1 (80.8 to 83.4)	−26.6 (−28.8 to −24.4)
Cantabria	84.6 (84.6 to 84.6)	15.2 (14.2 to 16.2)	Valencian Community	79.2 (78.3 to 80.2)	−13.7 (−15.7 to −11.8)
Asturias	81.4 (81.4 to 81.4)	−14.4 (−15.3 to −13.5)	Cantabria	78.4 (76.9 to 80.0)	39.0 (36.6 to 41.4)
Navarre	76.0 (76.0 to 76.0)	−14.4 (−15.4 to −13.4)	Canary Islands	67.2 (66.5 to 68.0)	−7.7 (−10.1 to −5.3)
La Rioja (reference category)	71.4 (71.4 to 71.4)	0.0 (. to .)	Asturias (reference category)	66.0 (66.0 to 66.0)	0.0 (. to .)

^a^ Listed in decreasing order of the volume of beer/wine purchases prior to the event.

**Table 3 nutrients-14-03412-t003:** Unstandardized coefficients from interrupted time series ARIMA regression models for intercept and level change associated with the introduction of 20% ABV variants of same-branded whisky or gin for 30 households that had purchased at least one bottle of the 20% ABV variants, controlling for changes in purchases by 1178 households that had purchased the regular-strength but not the 20% variants.

	Intercept ^a^	Level Change ^a^
Bottles (adjusted to 700 m per bottle) of 20% ABV variant	0.00	0.96 (0.95 to 0.97)
Bottles (adjusted to 700 m per bottle) of regular-strength variant	1.29 (1.28 to 1.30)	0.07 (0.05 to 0.10)
All grams of alcohol within all products	120.32 (118.12 to 122.52)	−8.70 (−15.76 to −1.65)
All grams of alcohol within all studied variants of branded whisky and gin	155.97 (155.76 to 156.17)	22.17 (21.51 to 22.82)
All grams of alcohol within all other spirits (excluding all studied variants of branded whisky and gin)	161.72 (161.40 to 162.05)	−41.92 (−42.97 to −40.87)
All grams of alcohol within all spirit products	158.80 (157.82 to 159.79)	−26.70 (−29.84 to −23.56)
Mean alcohol by volume (ABV) in all spirit products	39.36 (39.28 to 39.44)	−6.09 (−6.35 to −5.83)

^a^ The model includes, as independent variables, the event (time of introduction of 20% ABV variants), whether or not the household had purchased the 20% variant, and the interaction term event*whether or not.

## Data Availability

No additional data are available. Kantar WorldPanel data cannot be shared due to licensing restrictions.

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
