# Peer review of "The Impact of Lower-Strength Alcohol Products on Alcohol Purchases by Spanish Households"

_nutrients, 2022, doi:10.3390/nu14163412_

Round 1

Reviewer 1 Report

This article is a a significant contribution to the literature that addresses ways to reduce alcohol consumption, with the intent to reduce alcohol problems and alcoholism. The authors build on the important work of the World Health Organization and with their previous work on British alcohol purchase data, are ideally suited to expand their work into Spanish households.

I will comment on how to expand the Introduction/literature at the end of this review.

The 18,954 Spanish households with 1.29 million purchases is huge and remarkable!

I hope you have a research methodologist/statistician reviewing the article because I do not have this expertise. The three hypotheses are clear. I was tempted to suggest a simpler presentation of analyses and discussion but then looked back at the title of the journal, Nutrients. It seems evident that the article is written for the scientific community rather than the lay population and therefore I will not make that suggestion.

I suggest a simpler title of the article: "The impact of lower-strength alcohol products on alcohol purchases by Spanish households." The title as it now stands is too long and cumbersome.

Throughout the article, the authors state "less grams." In the U.S. it is considered more correct to state "fewer grams." It is the difference between items that are countable and those items that are not. For example, fewer people; less sugar.

Regarding the discussion section, I suggest removing the subheadings of: "main findings," "what is already known on this topic," "what this study adds," and "limitations of the study." These subheadings are simply not needed.

I would remove the long paragraph under "what is already known on this topic" currently the line #s 328-347, and place them at the beginning of the article to mesh them with line #s 48-53. This improves and adds to the literature review in the Introduction.

The first paragraph of the article alludes to WHO recommendations and the authors could add something about how effective they have been. For example, it is now well-known that higher alcohol prices and decreased availability strategies do work to reduce alcohol consumption. This bolsters the literature review a bit more.

Overall, a very good effort!

Author Response

Reviewer 1

I suggest a simpler title of the article: "The impact of lower-strength alcohol products on alcohol purchases by Spanish households." The title as it now stands is too long and cumbersome.

RESPONSE: Thank you for you helpful comments throughout. We have shortened the title.

Throughout the article, the authors state "less grams." In the U.S. it is considered more correct to state "fewer grams." It is the difference between items that are countable and those items that are not. For example, fewer people; less sugar.

RESPONSE: Yes, you are correct. Have changed throughout.

Regarding the discussion section, I suggest removing the subheadings of: "main findings," "what is already known on this topic," "what this study adds," and "limitations of the study." These subheadings are simply not needed.

RESPONSE: We have removed the headings.

I would remove the long paragraph under "what is already known on this topic" currently the line #s 328-347, and place them at the beginning of the article to mesh them with line #s 48-53. This improves and adds to the literature review in the Introduction.

RESPONSE: Thank you. We have done so.

The first paragraph of the article alludes to WHO recommendations and the authors could add something about how effective they have been. For example, it is now well-known that higher alcohol prices and decreased availability strategies do work to reduce alcohol consumption. This bolsters the literature review a bit more.

RESPONSE: Yes, have added this in.

Reviewer 2 Report

The article is particularly interesting in the context of establishing new targets regarding the reduction of alcohol consumption. Of course, completely giving up alcohol is the best attitude, but as a public health measure, the introduction of drinks with low or no alcohol content can be an effective way to reach the final goal. The authors investigate the impact of the presence on the market of some drinks with low alcohol content on the quantity of drinks bought and on the evolution of the grams of alcohol consumed in Spanish households. A decrease is postulated, both in terms of beer, as well as wine and spirits. The analysis confirms all three hypotheses, sustaining the usefulness of the presence on the market of drinks with reduced alcohol content. 

The scientific approach is correct, the tables and figures are suggestive and complete the understanding of the text. The bibliography is adequate.

Please revise line 232, because you make a reference to wine, although the paragraph refers to beer.

Author Response

Reviewer 2

Please revise line 232, because you make a reference to wine, although the paragraph refers to beer.

RESPONSE: Thank you for your comments. We have corrected this.